# Genome-Wide Identification, Characterization and Expression Analysis of Toll-like Receptors in Marbled Rockfish (*Sebastiscus marmoratus*)

**DOI:** 10.3390/ijms231911357

**Published:** 2022-09-26

**Authors:** Yuan Zhang, Xiaoyan Wang, Fei Han, Tianxiang Gao

**Affiliations:** 1Fisheries College, Ocean University of China, Qingdao 266003, China; 2National Engineering Research Center for Marine Aquaculture, Zhejiang Ocean University, Zhoushan 316002, China; 3Fishery College, Zhejiang Ocean University, Zhoushan 316022, China

**Keywords:** *Sebastiscus marmoratus*, toll-like receptor, genome-wide identification, poly(I:C), immune response

## Abstract

Toll-like receptor (TLR) is a cluster of type I transmembrane proteins that plays a role in innate immunity. Based on the marbled rockfish (*Sebastiscus marmoratus*) genome database, this study used bioinformatics methods to identify and analyze its TLR gene family members. The results showed that there were 11 TLR gene family members in *Sebastiscus marmoratus* (SmaTLR), which could be divided into five different subfamilies. The number of amino acids encoded by the Sma*tlr* genes ranged from 637 to 1206. The physicochemical properties of the encoded proteins of different members were also computed. The results of protein structure prediction, phylogenetic relation, and motif analysis showed that the structure and function of the SmaTLRs were relatively conserved. Quantitative Real-Time PCR (qRT-PCR) analysis revealed the expression patterns of SmaTLRs in the gill, liver, spleen, head kidney, kidney, and intestine. SmaTLRs were widely detected in the tested tissues, and they tended to be expressed higher in immune-related tissues. After polyriboinosinic polyribocytidylic acid (poly(I:C)) challenge, SmaTLR14, SmaTLR3, SmaTLR5S, SmaTLR7, and SmaTLR22 were significantly upregulated in the spleen or liver. The results of this study will help to understand the status of TLR gene family members of marbled rockfish and provide a basis for further study of the functional analysis of this gene family.

## 1. Introduction

Toll-like receptors (TLRs) are the major pattern recognition receptors (PRR) of the innate immune system in vertebrates, which can initiate host immune responses by recognizing potential pathogens. Innate immunity mediated by TLR is the first defense against pathogenic microorganisms in vertebrates. It plays an important role in bridging innate and acquired immune responses [1,2].

The first member of the TLR gene family was initially identified in the fruit fly (*Drosophila melanogaster*) [3]. This receptor was named Toll and was found to be responsible for the embryogenic dorsoventral development of fruit flies and protection against fungi in adult flies [4]. The TLR gene family is very conserved in mammals. Human beings have 10 members of the TLR gene family, and the mouse (*Mus musculus*) has 13 members [5,6]. Interestingly, whole genome and segmental duplications contribute to the expansion and functional diversification of TLR gene family members. For example, teleost has at least 22 functional TLR genes [7,8,9], including *TLR1*, *TLR2*, *TLR3*, *TLR5*, *TLR7*, *TLR8*, *TLR9*, and *TLR13*, which are common in the TLR gene family between fish and mammals [10,11]. At the same time, fish TLR genes exhibit unique evolutionary characteristics. For example, except for a few fish species, such as zebrafish (*Danio rerio*) and grass carp (*Ctenopharyngodon idella*), the *TLR4* gene is missing from most fish [12]. *TLR1*, *TLR6*, *TLR2*, and *TLR5* of Atlantic cod (*Gadus morhua*) are missing, while the *TLR7*, *TLR8*, *TLR9*, *TLR22*, and *TLR25* genes are significantly expanded. The expansion of TLRs in Atlantic cod might be a compensatory adaption to counterbalance other relevant alterations of the immune system [13].

Leucine-rich repeats (LRRs) encoded by the TLR gene compose an extracellular domain, which can specifically identify different types of pathogen-associated molecular patterns (PAMP), xenobiotic-associated molecular patterns (XAMPs), and danger-associated molecular patterns (DAMPs) [14,15]. In mammals, the stimulation of TLRs by their ligands triggers a signaling pathway common for all TLRs, resulting in the activation of myeloid differentiation factor 88 (MyD88), nuclear factor-κB (NF-κB), which in turn produces inflammatory cytokines (Figure 1). Members of the TLR1 subfamily (TLR1, TLR2, TLR6, and TLR10) recognize triacylated lipoprotein and peptidoglycan from bacteria when they form a heterodimer [16,17]. TLR3 can be activated by double-stranded RNA (dsRNA). TLR4 interacts with Gram-negative bacterial lipopolysaccharide (LPS) [18]. TLR5 is stimulated by bacterial flagellin [19]. TLR7 and TLR8 share similarities in structure, localization, and function, which are known to sense single-stranded RNA (ssRNA) from viruses [20]. TLR9 responds to unmethylated cytosine-phosphate-guanine (CpG) DNA [21]. Orthologs of mammalian TLRs in fish share common features with those in mammals when generating immune responses. Interestingly, some non-mammalian TLRs in fish exhibit specificity in ligand recognition. For example, both TLR9 and TLR21 are identified and found to recognize CpG DNA in fish, but they respond preferentially to CpG DNA with different CpG motifs [21]. In addition, fish-specific TLR22 recognizes long-sized dsRNA on the surface, while the mammalian ortholog TLR3 is expressed in the endosome membrane to recognize short-sized dsRNA [22]. After the recognition of the ligands, TLRs activate downstream signaling pathways and trigger immune responses through the Toll/interleukin-1 receptor (TIR) domain and TIR domain-containing cohesive molecules, such as the myeloid differentiation primary response gene (Myd88) and other TIR domain-containing adaptor proteins [23]. TLR signaling pathway activation results in the induction of inflammatory cytokines and costimulatory molecules as the host’s anti-bacterial or anti-viral immune responses [24].

Previous research on marbled rockfish mainly focused on biological characteristics, reproductive habits, physiological and ecological responses, and genetic characteristics. In recent years, studies on the immune function of marbled rockfish have gradually been carried out, such as the immune defense of marbled rockfish against fungal infection and parasitic infection [25,26], and the antimicrobial mechanism of its multiple antimicrobial peptides [27]. Therefore, it is of great significance to study and analyze the TLR gene family of marbled rockfish for a more comprehensive understanding of its immune system. Researchers have been focusing on the discovery of modulators of TLRs, which can act as immunotherapeutic or vaccine adjuvants for the treatment of cancer, allergies, and infectious diseases [15,28]. A double-stranded RNA virus is a kind of pathogen in aquaculture. Poly(I:C), as a synthetic double-stranded RNA, is a common vaccine adjuvant [29]. In many studies, fish have been injected with poly(I:C) as an alternative method for virus infection [30,31]. However, to date, dsRNA viruses and other viral mimics have not been examined experimentally or monitored in marbled rockfish. In this study, poly(I:C) was also used to simulate the challenge experiment of a double-stranded RNA virus on marbled rockfish.

In this study, we aim to utilize genome coding sequences of marbled rockfish for bioinformatics analysis of the marbled rockfish TLR gene family. We obtained 11 TLRs in marbled rockfish (termed SmaTLR) through whole-genome identification. Phylogenetic analysis, protein structure prediction, and analysis of conserved motifs were performed. Concomitantly, the expression patterns of SmaTLRs in different tissues (gill, liver, spleen, head kidney, kidney, and intestine) were detected. Additionally, the expression changes of SmaTLRs in the spleen and head kidney after the poly(I:C) challenge were investigated. The results of the present study would provide useful information for understanding the TLR gene family in marbled rockfish. The expression analysis should enrich the theoretical basis for selecting potential adjuvants and further studies on conservation approaches in marbled rockfish.

## 2. Results

### 2.1. Identification and Characterization of SmaTLRs in Marbled Rockfish

After local BLAST identification and bioinformatics analysis, a total of 11 different TLR genes were identified from the genomic database of marbled rockfish. These sequences were submitted to NCBI with accession numbers OM891535.1 to OM891545.1. The basic characteristics of the SmaTLR genes are shown in Table 1. Open reading frames (ORFs) were 1914 to 3621 bp in length, encoding 637 to 1206 amino acids with predicted molecular weights ranging from 71.43 to 134.59 kDa, and the theoretical pI varied from 5.94 to 9.

The 11 SmaTLR genes were distributed among 9 of 24 chromosomes in the marbled rockfish genome (Figure 2). SmaTLR2-2 and SmaTLR3 were both located on chromosome 3. SmaTLR14 and SmaTLR21 were both located on chromosome 11.

### 2.2. Phylogenetic Relation of the TLR Gene Family among Several Species

To investigate the phylogenetic relationship between SmaTLRs and the orthologs in other vertebrates, a phylogenetic tree was constructed based on the amino acid sequences of SmaTLRs and other species (Figure 3). The 11 SmaTLRs were clustered into 5 subfamilies according to the rules of classification used in other species. This results provided phylogenetic evidence of SmaTLRs. In marbled rockfish, SmaTLR1-1, SmaTLR1-2, SmaTLR2-1, SmaTLR2-2, and SmaTLR14 were clustered into the TLR1 subfamily, which included the largest number of TLR genes. Compared to other species, TLR6, TLR10, and TLR14 were absent in the TLR1 subfamily in marbled rockfish, while there were 5 members of the TLR11 subfamily. According to the result of alignment to the protein database, the multiple copies were annotated as “SmaTLR1-1”, “SmaTLR1-2”, “SmaTLR2-1”, “SmaTLR2-2”, and “SmaTLR14”, following the orthologous genes in other species. As shown in Figure 3, most teleosts exhibited an absence of TLR4, including marbled rockfish. The TLR3 subfamily, TLR5 subfamily, and TLR7 subfamily consisted of only one member in each subfamily. The members of the TLR11 subfamily included SmaTLR13, SmaTLR21, and SmaTLR22.

### 2.3. Gene Structure Characterization and Protein Domain Prediction

The prediction of protein domains showed that the proteins encoded by the SmaTLR genes were mainly composed of three functional regions: the extracellular region, transmembrane region (TM), and intracellular region (Figure 4). The extracellular region consisted of 4–15 LRRs. Among the 11 SmaTLRs, SmaTLR5S possessed only LRRs and lacked the TIR domain and a transmembrane region. TLR7 had no transmembrane region.

As shown in Figure 5, 20 different kinds of conserved motifs in the legend were arranged according to the number of occurrences in SmaTLRs. Motifs 13, 3, and 5 were widely distributed in the 11 SmaTLR protein sequences. The distribution of motifs in SmaTLRs from the same subfamily was highly similar. The type and number of motifs between SmaTLR2-1 and SmaTLR2-2, and SmaTLR21 and SmaTLR22 showed high similarity, indicating that the function of SmaTLRs may be conserved within the same subfamily. Comparing the TIR domain of SmaTLRs, there were three highly conserved regions: BOX1, BOX2, and BOX3. Except for SmaTLR5S, the 10 TLRs of marbled rockfish were highly conserved in the corresponding regions (Figure 6).

### 2.4. Expression of SmaTLRs in Different Tissues

To investigate the expression patterns of SmaTLRs, qRT-PCR was applied to examine the mRNA expression levels of SmaTLRs in different tissues in marbled rockfish (Figure 7). The results showed that SmaTLRs were widely expressed in the gills, liver, spleen, head kidney, kidney, and intestine in marbled rockfish. Most of the SmaTLRs were highly expressed in immune-related tissues (i.e., spleen, head kidney, liver, and gill), with a relatively low distribution level in other tissues. Among these, the highest expression levels of SmaTLR1-1, SmaTLR2-1, SmaTLR21, and SmaTLR22 were detected in the spleen. SmaTLR7 and SmaTLR14 were significantly expressed in the head kidney. SmaTLR1-2 and SmaTLR13 exhibited high expression levels in the gill. SmaTLR2-2, SmaTLR3, and SmaTLR5S were highly expressed in the liver. For SmaTLR1-1 and SmaTLR7, the expression levels were lowest in the gill. Relatively lower expression levels of SmaTLR14 were observed in the gill, liver, and spleen.

### 2.5. Expression of SmaTLRs after Poly(I:C) Challenge

To further explore the expression of SmaTLRs after the poly(I:C) challenge, the expression levels of mRNA in the spleen and head kidney of marbled rockfish were measured by qRT-PCR. As depicted in Figure 8a,b, the 11 SmaTLRs showed different expression changes after poly(I:C) injection.

In the spleen, 5 of the 11 detected genes were upregulated (SmaTLR2-1, SmaTLR3, SmaTLR5S, SmaTLR7, and SmaTLR14) after poly (I:C) injection, while the other 6 SmaTLRs were downregulated immediately (SmaTLR1-1, SmaTLR1-2, SmaTLR2-2, SmaTLR13, SmaTLR21, and SmaTLR22). Among them, SmaTLR3 (*p* < 0.05) and SmaTLR7 (*p* < 0.01) were significantly upregulated at 6 hpi, while SmaTLR1-1 was significantly downregulated at 6 hpi (*p* < 0.05). SmaTLR1-1 and SmaTLR1-2 had similar expression patterns, both showing downregulation at 6, 12, and 48 hpi compared with the control group, and upregulation at 24 hpi. SmaTLR1-2 was significantly upregulated at 24 hpi (*p* < 0.01). The expressions of SmaTLR2-1 and SmaTLR14 were continuously upregulated at each detected time point after injection. The expression of SmaTLR2-1 exhibited significant upregulation at 24 hpi (*p* < 0.05). SmaTLR14 was significantly upregulated at 12 and 24 hpi (*p* < 0.01). SmaTLR2-2 and SmaTLR13 were downregulated at first and then upregulated. Both showed significant upregulation at 48 h (SmaTLR2-2 *p* < 0.05; SmaTLR13 *p* < 0.01).

In the head kidney, 6 of the 11 tested genes were upregulated (SmaTLR2-2, SmaTLR3, SmaTLR5S, SmaTLR7, SmaTLR14, and SmaTLR22) after poly(I:C) injection, while the expression of the other 5 SmaTLRs (SmaTLR1-1, SmaTLR1-2, SmaTLR2-1, SmaTLR13, and SmaTLR21) were downregulated at the first time point. SmaTLR1-1 was downregulated at first, and then significantly upregulated from 24 hpi (*p* < 0.01). The expression of SmaTLR2-1 decreased continuously, and significant expression changes happened at 6 and 48 hpi compared with the control group (*p* < 0.01). A similar expression pattern was detected in SmaTLR3 and SmaTLR5S, which were upregulated continuously at 6, 12, and 24 hpi (*p* < 0.01), but downregulated at 48 hpi. SmaTLR14 was firstly upregulated at 6 and 12 hpi (12 hpi *p* < 0.01), and then downregulated at 24 and 48 hpi. The expression of SmaTLR22 continuously increased after poly (I:C) stimulation, showing extremely significant upregulation at 24 and 48 hpi (*p* < 0.01).

## 3. Discussion

In this study, the TLR gene family was identified from the whole genome coding sequence of marbled rockfish. We explored the nucleic acid sequences of the 11 SmaTLRs, the protein structure encoded by the genes, the expression and distribution of the genes in different tissues, and the expression changes of each SmaTLR under the poly (I:C) challenge.

### 3.1. Characterization of TLRs in Marbled Rockfish

The 11 SmaTLRs, together with the homologs of other species, fell into evolutionary branches of different subfamilies in the phylogenetic tree, which were very conservative in evolution. The TLR4 subfamily was absent in marbled rockfish, while there were 2 copies of SmaTLR1 and SmaTLR2, respectively. TLR4 was identified in only a few fish species, including zebrafish, common carp (*Cyprinus carpio*), grass carp, rare minnow (*Gobiocypris rarus*), and channel catfish (*Ietalurus punetaus*) [32,33]. Mammalian TLR4 recognized bacterial LPS, but stimulation with LPS in zebrafish did not induce the expression of TLR4 [34,35], indicating that there are some differences in the function of TLR4 between mammals and fish. Two types of TLR5 were found in teleosts: membrane-bound TLR5M and soluble form TLR5S. SmaTLR5S only had the LRR domain; this was the main difference between TLR5M and TLR5S [36]. TLR5S without the TIR domain or transmembrane region was also found in black rockfish [37], Atlantic salmon (*Salmon salar*) [38], Japanese flounder (*Paralichthys olivaceus*) [10], etc. Teleost could encode most of the TLR homologous proteins found in mammals, birds, and amphibians (TLR1~5, TLR7~9, and TLR12~13), as well as more than ten kinds of fish-specific TLRs (TLR14, TLR18~28) [8,9,39]. In cephalochordates and echinoderms, the expansion and diversity of TLRs are regarded as reflecting the evolution of a gene family to cope with a wide variety of microorganisms or pathogens in the aquatic environment [40,41,42]. The diversity of TLRs in marbled rockfish is supposed to be of great significance to their resistance to pathogens in their living waters.

The SmaTLR family genes encoded 637–1206 amino acids. The length of the protein sequence was similar to that of TLRs in other fish [43,44]. The SmaTLR proteins basically consist of three typical functional regions: an extracellular domain, a transmembrane domain, and an intracellular domain. The transmembrane domain is cysteine rich. The extracellular domains usually have 2–25 LRRs, each of which contains a 20–30-amino acid sequence that includes the LXXLXLXXN motif [45]. X can be any amino acid; L (leucine) can also be occupied by hydrophobic residues, including valine, isoleucine, and phenylalanine; and N (asparagine) can be occupied by cysteine, threonine, and serine [46,47]. LRRs are involved in signal transduction [48] and are also responsible for the recognition of PAMPs from bacteria and parasites, as well as fungi, and viruses [49]. From an evolutionary point of view, LRRs were selected by evolution to adapt to the changes in PAMPs and evolve accordingly, so they were typical of great variability. On the other hand, the TIR domain is highly conservative of all TLRs [50,51]. The TIR domain has highly homologous sequences with the cytoplasmic domain sequence of the interleukin-1 receptor 1L-1R family [24]. Besides recruiting adaptor proteins, TIR can also activate downstream signaling pathways, and trigger intracellular immune responses [52]. Most of the homology of TIR domain-containing TLR family members is confined to three conserved functionally important motifs: BOX1 ((F/Y) DAFISY), BOX2 (GYKLC—RD—PG), and BOX3 (a conserved W surrounded by basic residues) [53], in which X can be any amino acid; F (phenylalanine) can also be occupied by Y (tyrosine). In this study, the TLR proteins of marbled rockfish retained the characteristic motifs in the TIR domain. These motifs in the TIR domains of TLRs are involved in critical functions, such as interacting with the TIR domain of MyD88 to initiate downstream signaling cascades (BOX1 and BOX2) [54], as well as controlling the subcellular location of the receptor (BOX3) [53]. The proline (P) in BOX2 is a conservative amino acid with an assistant recognition function. This amino acid has been found conservative in TLR21 of chickens (*Gallus gallus*), rats (*Rattus norvegicus*), and fugu (*Fugu rubripes*) [55,56]. The amino acid at this site in BOX2 of SmaTLR21 is L (leucine). The results of multiple sequence alignment of zebrafish TLR21 and large yellow croaker TLR21 showed the same situation [57], indicating that the change in the conservative amino acid of TLR21 in fish was not accidental in marbled rockfish. However, further studies are required to clarify the effects of these changes on TIR function.

The analysis of the gene structure, protein structure, and species specificity of Toll-like receptors in marbled rockfish can provide a theoretical basis for the identification of pathogens and the development of the design of safe and efficient vaccines or medicines.

### 3.2. Expression of SmaTLRs in Different Tissues of Healthy Individuals

The 11 SmaTLRs were widely detected in different tissues. SmaTLR1-1, SmaTLR 1-2, SmaTLR 2-1, SmaTLR2-2, and SmaTLR14 of the TLR1 subfamily presented high expression levels in immune-related tissues. TLR1 subfamily members were also highly expressed in immune-related organs, such as the spleen and kidney, in other fish, including Japanese meagre (*Argyrosomus japonicus*), Nile tilapia (*Oreochromis niloticus*), Japanese flounder, and channel catfish [8,58,59,60]. In marbled rockfish, the highest expression levels of SmaTLR3 and SmaTLR5S were found in the liver. TLR5S was also found to be highly expressed in the livers of various fish, including channel catfish, mandarin fish (*Siniperca chuatsi*), rainbow trout (*Oncorhynchus mykiss*), and Japanese flounder [60,61,62,63], while TLR3 and TLR5S of Japanese meagre were mainly detected in the spleen [58]. SmaTLR7 showed the highest expression level in the kidneys. Similar expression patterns of TLR7 have been reported in black rockfish and mandarin fish [37,63]. SmaTLR13 was mainly expressed in the gill. SmaTLR21 and SmaTLR22 were mostly expressed in the spleen. In other fish, TLR13 and TLR21 also showed high expression levels in immune-related tissues. For example, Nile tilapia TLR13a and TLR13b were significantly highly expressed in the spleen, and TLR21 in the lined seahorse (*Hippocampus erectus*) was mostly detected in its kidney and gill [64,65]. It was observed that TLR22-1 and TLR22-2 of common carp, and TLR22 of black rockfish were highly expressed in the spleen [32,37]. In Japanese flounder, TLR22 showed high expression levels in the spleen, liver, pituitary, and gill [60]. These findings indicate that TLR22 may be more widely involved in the immune response in immune-related tissues than other TLR family members.

According to the existing research, TLRs had different tissue-specific expression patterns in different species, but some TLRs showed the highest expression in immune-related organs. Combined with the present study, the 11 SmaTLR genes were widely expressed in the tested tissues of healthy individuals, and some of them showed significantly high expression levels in the immune-related organs. It can be speculated that these SmaTLR genes participate in individual immune activity in immune organs.

### 3.3. Immune Response of SmaTLRs against Poly (I:C) Challenge

The molecular structure of SmaTLRs has vital significance in immunological function. It has been demonstrated that TLR activation is an important aspect of adjuvants in vaccines [66]. Therefore, we applied poly(I:C) as a virus analog to simulate viral infection, helping us learn the antiviral immune response of TLRs in marbled rockfish.

Although TLR1 subfamily members were highly expressed in the immune-related tissues of marbled rockfish, the expression regulation of each gene differed against the poly(I:C) challenge. The expression of SmaTLR14 in the spleen and head kidney was upregulated immediately after poly(I:C) injection, and the upregulation of SmaTLR14 in the spleen was continuous within 48 h after injection. In research on mandarin fish, a significant and continuous upregulation of TLR14 was also observed in the spleen after stimulation by poly(I:C) [63], indicating that TLR14 may actively participate in the immune response in fish spleen.

In the spleen and head kidney of marbled rockfish, the expression of TLR3 was significantly upregulated after poly(I:C) injection. TLR3 was also upregulated under poly(I:C) stimulation in the spleen of large yellow croaker, zebrafish, and mandarin fish [63,67,68], and in the head kidney of Japanese meagre and black rockfish [37,58]. In the virus infection experiment of rainbow trout and rare minnow, TLR3 expression was significantly upregulated in immune tissues [69,70]. These results indicated that the function of TLR3 in fish was similar to that in mammals, as it could produce immune responses to viruses and virus analogs [71].

TLR5M in Japanese flounder, rainbow trout, channel catfish, and other fish are homologous with TLR5 in mammals. TLR5S is unique to fish [7] and can produce an immune response under the stimulation of poly(I:C), LPS, and flagellin [72]. The expression of SmaTLR5S is not high in tissues other than the liver, but its expression changes significantly after being stimulated by poly(I:C). Research on *Cryptocaryon irritans* infected marbled rockfish indicated that TLR5 was the only activated TLR family member in immune-related tissues (spleen, head kidney, and liver) [26]. These results suggest that SmaTLR5S may play an important role in the immunity of marbled rockfish.

SmaTLR7 was highly expressed in the head kidney. After poly(I:C) injection, SmaTLR7 was immediately upregulated in the head kidney and spleen. TLR7 has been reported to recognize viral RNA and activate the antiviral immune mechanism in fish [62,71]. In the studies of large yellow croaker, Japanese meagre and black rockfish, TLR7 showed a similar regulation tendency with SmaTLR7 in the head kidney or spleen after poly(I:C) infection [37,58,73], implying that the virus analogues could also cause the immune response of fish TLR7.

SmaTLR22 displayed the highest expression level in the spleen, but the expression was not significantly changed under the poly(I:C) challenge. However, SmaTLR22 expression was continuously upregulated in the head kidney after poly(I:C) injection, showing extremely significant changes at 24 h and 48 h. TLR22 is a unique TLR gene for fish that is located on the cell membrane. It not only recognizes long-chain dsRNA but also reacts to aquatic bacteria and other PAMPs [22,60]. The expression of TLR22 was significantly upregulated after poly(I:C) injection in the head kidney of black rockfish and Japanese meagre, as well as in the spleen of mandarin fish and in the brood pouch of lined seahorse [37,58,63,74]. Despite differences in different tissues, it appears that TLR22 actively participates in the antiviral immune response in fish.

Taken together, this study provides a preliminary understanding of the TLR gene family in marbled rockfish. Exploring the characterization and expression pattern of SmaTLRs will be helpful in comprehensively understanding immune function after stimulation, and it is also of great significance to disease prevention in marbled rockfish.

## 4. Materials and Methods

### 4.1. Ethics Statement

All animal experiments were conducted in accordance with the guidelines of the Animal Research and Ethics Committee of the Ocean University of China. This study did not involve any protected or endangered animals. The fish were anesthetized by immersion in eugenol before sampling the fish tissues.

### 4.2. Identification of Members of the TLR Gene Family

To identify TLR family members in marbled rockfish, homologous sequences were downloaded as queries from the National Center for Biotechnology Information (NCBI). The genome coding sequences of marbled rockfish were obtained from the genome sequencing project of our lab (not openly available yet), in which the genome was sequenced by combing Illumina short-read sequencing, PacBio long-read sequencing (×145) and Hi-C sequencing. The genomic assembly was of high quality, with a scaffold N50 at 34.95 Mb, and the final assembly contained 96% complete BUSCOs. First, the TBLASTN of local BLAST 2.3.0+ was used to search for TLR genes in the marbled rockfish genome coding sequences with a cutoff *E-value* of 1 × 10^−5^. The sequences of candidate TLR family members in marbled rockfish were obtained and further identified by comparing them to the NCBI nonredundant (NR) protein sequence database [58].

### 4.3. Gene Structure Characterization and Protein-Conserved Domain Prediction

The gene characterizations of the SmTLR genes were calculated by the ProtParam tool of Expasy (https://web.expasy.org/protparam/ (accessed on 1 July 2020)). The open reading frame (ORF) was obtained via an ORF finder on NCBI (https://www.ncbi.nlm.nih.gov/orffinder/ (accessed on 6 July 2022)). Conserved motifs were identified using the MEME suite 5.4.1 online tool (http://meme-suite.org/tools/meme (accessed on 18 January 2022)) [75]. The conserved domains were predicted by the Normal mold in the online tool Simple Modular Architecture Research Tool (SMART, http://smart.embl-heidelberg.de/ (accessed on 26 October 2020)) with default parameters. Multiple sequence alignment was performed via DNAman 8.0.

### 4.4. Phylogenetic and Syntenic Analysis of TLR Genes in Marbled Rockfish

The phylogenetic tree was constructed based on the amino acid sequences of SmaTLRs and orthologous genes in other representative vertebrates, including zebrafish, large yellow croaker (*Larimichthy crocea*), miiuy croaker (*Miichthys miiuy*), orange-spotted grouper (*Epinephelus coioides*), Japanese pufferfish (*Takifugu rubripes*), black rockfish (*Sebastes schlegelii*) and human (*Homo sapiens*). Multiple alignments were conducted on all the amino acid sequences using the MUSCLE program in MEGA 6.0, and the tree was constructed using the neighbor-joining (NJ) method in MEGA 6.0 with 1000 bootstrap replications. The information on the sequences used to create the phylogenetic tree is shown in Appendix A.

### 4.5. Chromosomal Locations of TLR Genes in Marbled Rockfish

The gene position of the SmTLR genes was annotated in the marbled rockfish genome database. The chromosome distribution map of the SmTLR genes was depicted with MapChart 3.0 [76].

### 4.6. Challenge Experiment and Sample Collection

Marbled rockfish (average weight of 50 ± 5 g) were reared in a raising factory on Xixuan Island, Zhejiang Province. The marbled rockfish were acclimatized for 2 weeks before the experiment (temperature of 25 °C, salinity of 24, abundant oxygen) and fed daily with shrimps.

In order to examine the expression pattern of SmaTLRs in different tissues, six tissues were collected from 3 healthy individuals, including the gill, liver, spleen, kidney, head kidney, and intestine. The samples were dissected and immediately frozen in liquid nitrogen and stored at −80 °C refrigerator afterward.

To characterize the immune responses of the SmaTLRs, a poly(I:C) challenge experiment was conducted. Fish were divided into treated and control groups, and each group contained 16 fish (12 for the experiment and the rest for supplementary use). 200 μL of poly(I:C) at a concentration of 0.5 μg/mL was intraperitoneally injected into 16 individuals of the treated group. Another 16 marbled rockfish were injected intraperitoneally with 200 μL PBS as the control group. Spleens and head kidneys were collected in both the control and treated groups at 6, 12, 24, and 48 h after the injection. For each time point, a total of 3 individuals were randomly collected as 3 replicates (*n* = 3) from each group. Samples were immediately frozen in liquid nitrogen and stored at −80 °C refrigerator for RNA extraction.

### 4.7. RNA Extraction, cDNA Synthesis and qRT-PCR Analysis of SmaTLR Expression

Total RNA was extracted from the tissues collected from healthy fish and injected fish from the challenge experiment using an RNA Isolator (TRIzol) (Vazyme R401-01, Nanjing, China). The extracted RNA samples were detected by a microplate reader to determine concentration and purity. The A260/280 ratio should be between 1.8 and 2.0. Next, cDNA was synthesized using HiScript II-RT SuperMix for qPCR (+gDNA wiper) (Vazyme R223-01, Nanjing, China) according to the protocol. Then, the cDNA samples were stored at −20 °C for further experiments.

Primers used for qRT-PCR analysis were designed using Primer 5 software. β-actin was used as a suitable internal reference. Information on the primers used in this study is shown in Table 2. qRT-PCR was conducted on a StepOne Plus Real-Time PCR system (Applied Biosystem, Foster City, CA, USA) using TaKaRa TB Green Premix Ex Taq (Til RAaseH Plus, RR420A). The reaction was performed in a total volume of 20 μL, including 2 μL of cDNA (diluted at a 1/40 ratio), 0.4 μL of each primer (10 μM), 0.4 μL of ROX Reference Dye (50×), 10 μL of SYBR^®^ Premix Ex TaqTM II (Tli RNaseH Plus) (2×), and 6.8 μL of RNase-free water. The qRT-PCR cycling conditions were the 30 s at 95 °C followed by 40 cycles of 5 s at 95 °C, 30 s at 60 °C, 30 s at 72 °C, and one cycle of 15 s at 95 °C.

The number of threshold cycles (CT values) was collected, and the expression level of the target genes was calculated using the 2^−ΔΔCT^ method [77]. For the expression pattern of SmaTLRs in different tissues, the relative expression levels of each gene in different tissues were compared to those in the intestines. The relative expression level was represented by comparing the gene expression after the poly(I:C) challenge to the PBS control at the same time point and then calculating the log2-fold change in the ratio [78]. The variance of SmaTLR expression levels between the control and the treated groups was determined by SPSS 13.0. The results were detected using one-way ANOVA, and statistical significance was defined at *p* < 0.05.

## 5. Conclusions

In this study, the 11 members of the TLR gene family were identified from the whole-genome coding sequences of marbled rockfish. Phylogenetic analysis showed that SmaTLRs shared high homology with their homologs from other teleost species. Protein structure prediction and analysis of conserved motifs suggested that the SmaTLRs in marbled rockfish were highly conserved. In addition, this study provides information for further studies on immunology related to the TLR family in marbled rockfish and other Scorpaenidae species, which would make sense in vaccine design and resource conservation of these species. The distribution of SmaTLRs was widely detected in different tissues, and most SmaTLRs showed higher expression levels in immune-related tissues. After poly(I:C) challenge, the 11 SmaTLRs had a different regulation tendency. SmaTLR14, SmaTLR3, SmaTLR5S, SmaTLR7, and SmaTLR22 were significantly upregulated in the spleen or livers of marbled rockfish. Above all, these results provided a comprehensive understanding of the SmaTLR gene family and enriched the theoretical basis for further study of efficient fish vaccines or adjuvants.

## Figures and Tables

**Figure 1 ijms-23-11357-f001:**
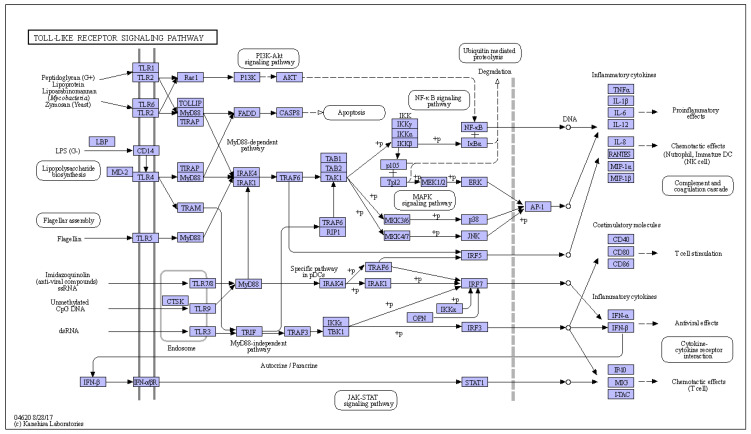
Overview of the toll-like receptor signaling pathway (https://www.kegg.jp/pathway/ko04620 (accessed on 28 August 2017)).

**Figure 2 ijms-23-11357-f002:**
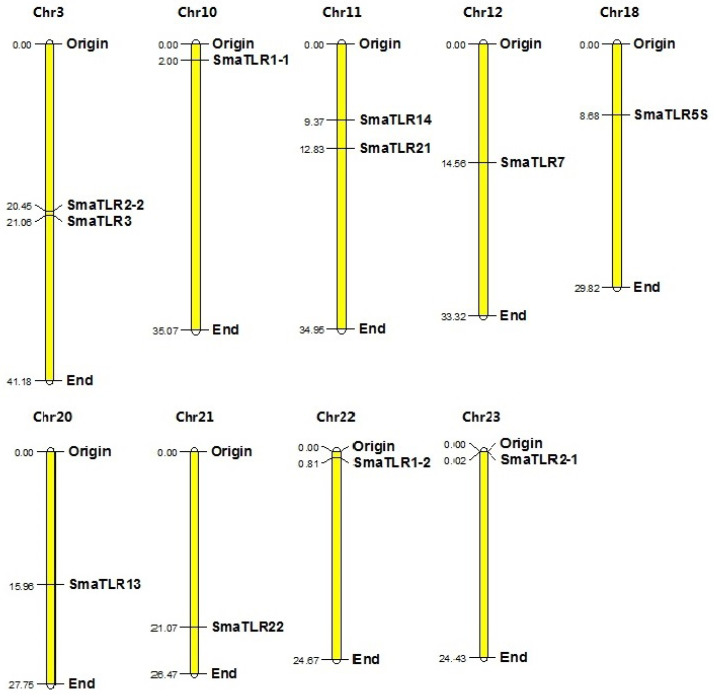
Chromosomal locations of the 11 TLRs in marbled rockfish. The origin and end represent the starting and ending positions of the chromosomes, respectively. The numbers in the figure represent the length of the chromosomes and the position of the genes. The unit is Mb.

**Figure 3 ijms-23-11357-f003:**
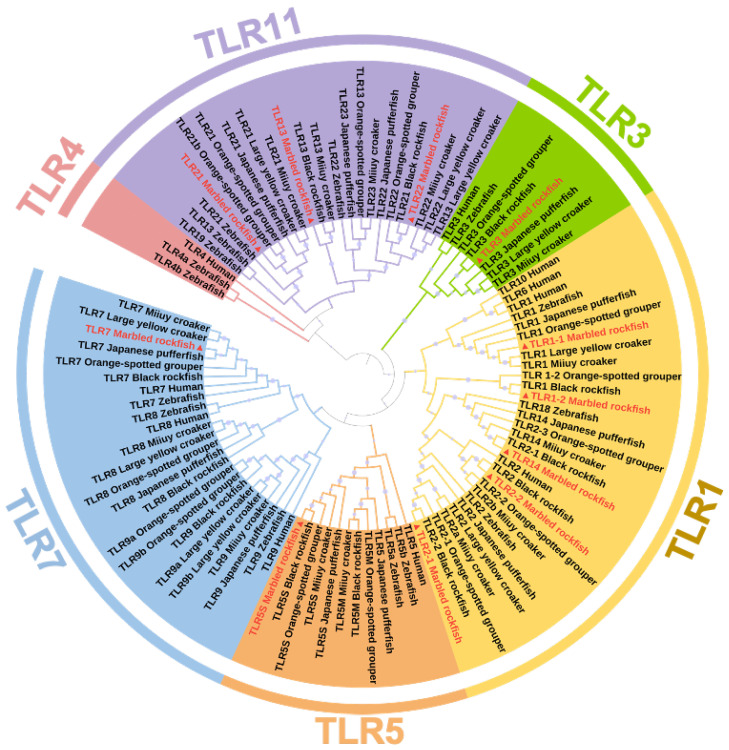
Phylogenetic tree of TLR gene families of selected species. TLRs from marbled rockfish are marked with triangle symbols and red font. The percentage of replicate trees in which the associated taxa clustered together in the bootstrap test (1000 replicates) is shown by spot symbols of different sizes on the branches.

**Figure 4 ijms-23-11357-f004:**
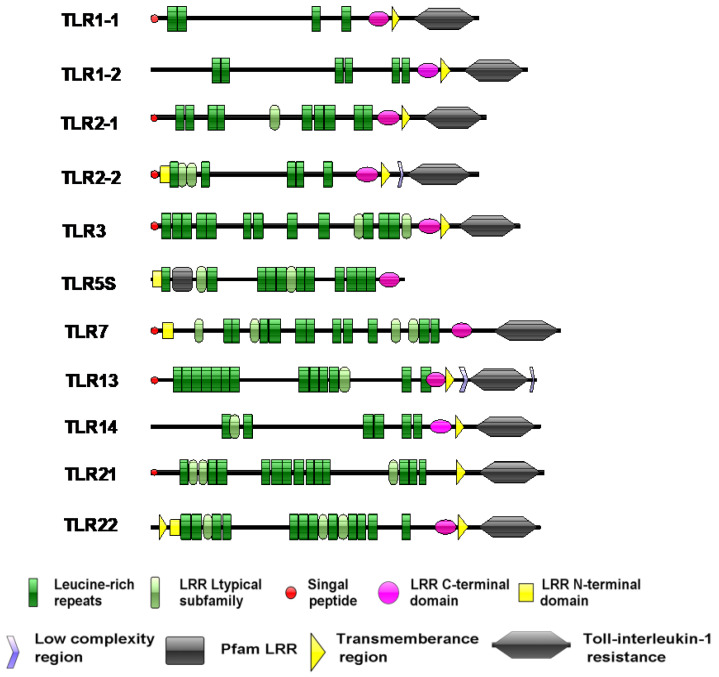
The structure features prediction of TLRs in marbled rockfish. Green hexagonals represent TIR; green bars represent LRRs; blue bars represent transmembrane region (TM); red bars represent signal peptide.

**Figure 5 ijms-23-11357-f005:**
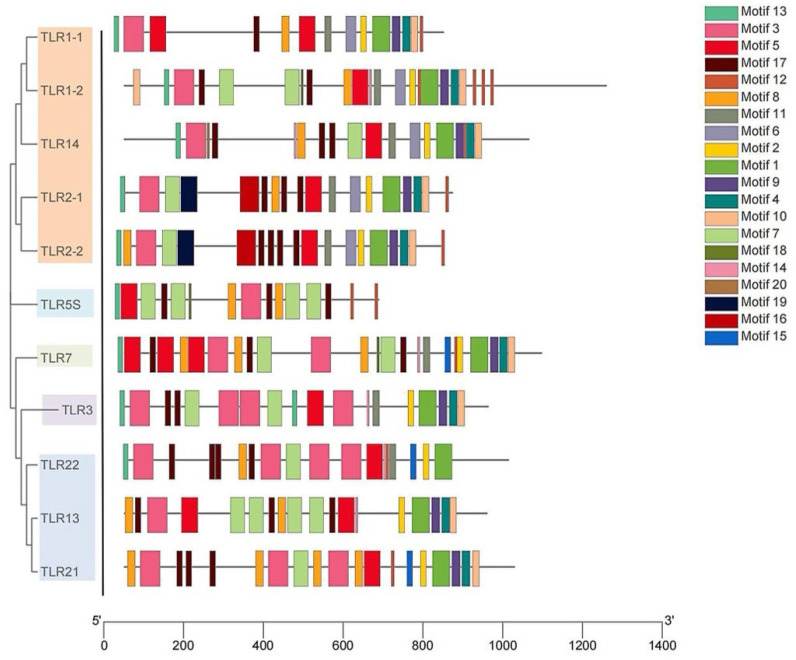
Conserved motifs analysis of TLR genes in marbled rockfish.

**Figure 6 ijms-23-11357-f006:**
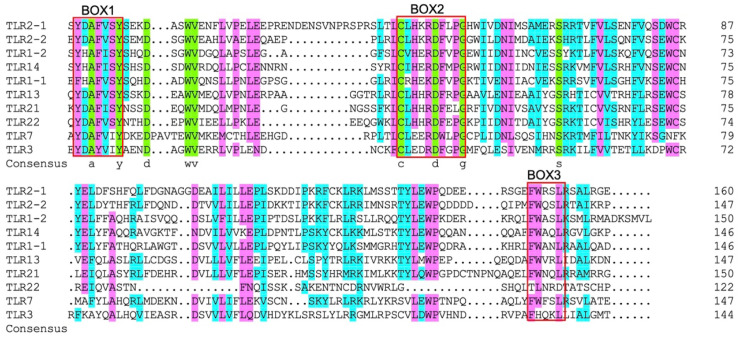
Multiple sequence alignment of the TIR domain in TLRs in marbled rockfish.

**Figure 7 ijms-23-11357-f007:**
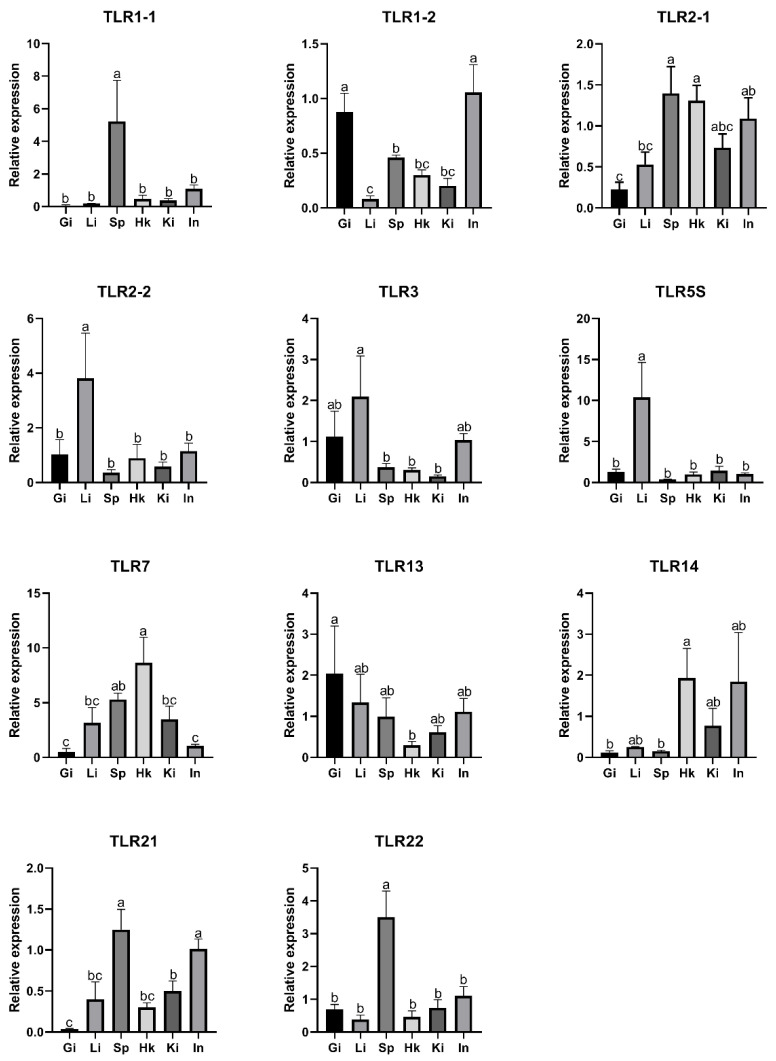
Expression of TLRs in selected tissues in marbled rockfish. The relative gene expression levels of SmaTLRs were normalized with β-actin. Different letters above the error bars indicated statistical significance (Duncan’s test, *p* < 0.05). Gi: gill; Li: liver; Sp: spleen; Hk: head kidney; Ki: kidney; In: intestine.

**Figure 8 ijms-23-11357-f008:**
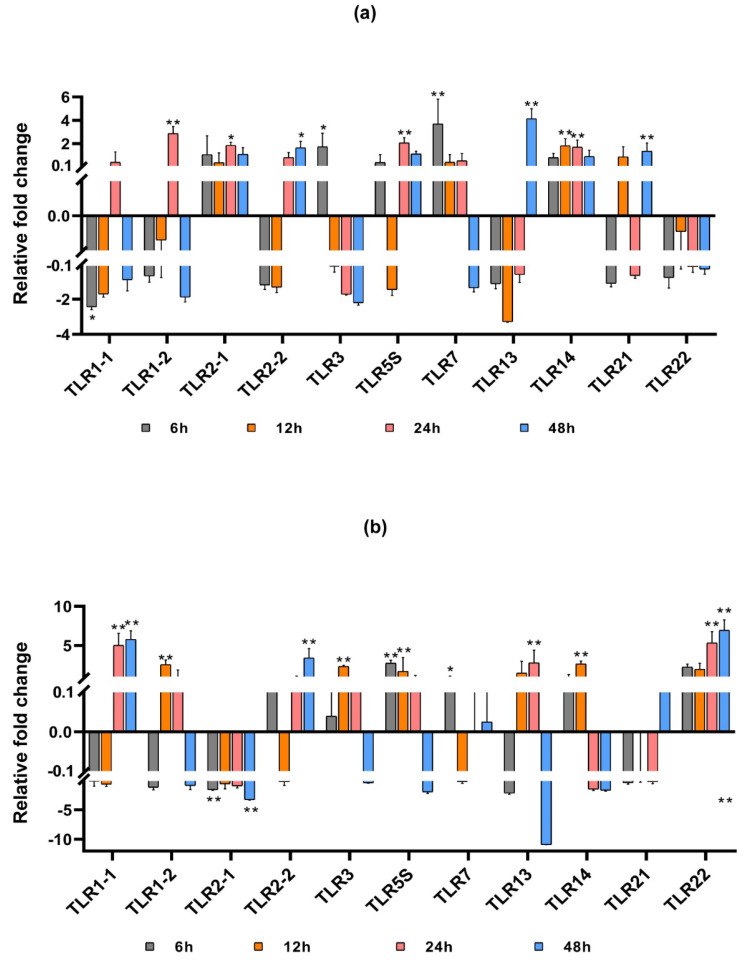
Expression profiles of TLRs in the spleen (**a**) and head kidney (**b**) of marbled rockfish after poly(I:C) injection. The relative gene expression levels of SmaTLRs were normalized with β-actin. Asterisks indicated statistical significance (*p* < 0.05, *) or extremely statistical significance (*p* < 0.01, **) of up-/downregulation of the genes between the treated group and the control at the given time point.

**Table 1 ijms-23-11357-t001:** Summary of TLR genes identified in marbled rockfish genome.

Name	Gene ID	ORF (bp)	Amino Acid	MW (kDa)	pI	Chr	Accession Number
TLR1-1	Seb006043	2400	799	90.40	6.21	chr10	OM891535.1
TLR1-2	Seb020825	3621	1206	134.59	6.5	chr22	OM891536.1
TLR2-1	Seb022133	2466	821	93.23	6.21	chr23	OM891537.1
TLR2-2	Seb021445	2397	798	90.37	5.94	chr3	OM891538.1
TLR3	Seb019353	2733	910	103.49	8.85	chr3	OM891539.1
TLR5S	Seb003500	1914	637	71.43	8.21	chr18	OM891540.1
TLR7	Seb001894	3135	1044	119.93	8.26	chr12	OM891541.1
TLR13	Seb002611	2724	907	102.44	8.66	chr20	OM891542.1
TLR14	Seb013611	3039	1012	116.55	8.62	chr11	OM891543.1
TLR21	Seb016897	2931	976	113.19	8.41	chr11	OM891544.1
TLR22	Seb013813	2886	961	109.25	9	chr21	OM891545.1

**Table 2 ijms-23-11357-t002:** Primers used in this experiment.

Gene	Forward Primer (5′-3′)	Reverse Primer (5′-3′)
*TLR1-2*	CTCCTATTGGTTCGTCACTT	TTCCTGCCTCTCCTCTTC
*TLR2-1*	TGCGATACCTCAACATCTC	GCTCTCCTCTCCATCTGT
*TLR2-2*	GCGTTCAGCGGTAATAATC	CCAGGTCCAGGTCATCTT
*TLR3*	CTACAACCTCCTCAATAGTCT	CAGTGCTGCTCAGTATGT
*TLR5S*	CGCTTCAAGACTCCATCC	CCAACTAGATCAGACTCACAT
*TLR7*	CACATACAGCACATTGAGAA	TAGAGGATGATTGACAGGTAG
*TLR13*	CTTACCTCAGTCGCCATC	GCTCTTCTGTCGTTGTCA
*TLR14*	TCACGCCACACCAATAAC	GATGTCAAGATGCTCCAGTA
*TLR21*	TCACCATCTCCTCTAATCCT	CACAGACTCAGCAATCACT
*TLR22*	GCCTTCGTCTCCTACAAC	GATGGTCTTCCTGCTTCC
*β-actin*	AGGGAAATCGTGCGTG	ATGATGCTGTTGTAGGTGGT

## Data Availability

The sequences of SmaTLRs were submitted to NCBI with accession numbers OM891535.1 to OM891545.1.

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
