# Peer review of "Genome-Wide Identification, Characterization and Expression Analysis of Toll-like Receptors in Marbled Rockfish (*Sebastiscus marmoratus*)"

_ijms, 2022, doi:10.3390/ijms231911357_

Round 1

Reviewer 1 Report

Abstract

This section is well prepared; I recommend the authors to include more detail in protein information of TLR family.

Introduction

This section is well prepared.

Results

This section is well prepared.

Discussion

This section is well prepared, and I would like to ask some thought-provoking questions to the authors.

The authors indicate that this TLR family is well represented in juvenile marble rocky fish. How would the genomic expression of this family be modified in early life stages, particularly since it may have another function during early ontogeny?

Would it be possible to include TRL family changes during the larval period?

It is clear that after Poly(I:C) injection, there is a generalized increase in several of the TRL subfamilies. How could the results be modified if a pathogen (virus or bacterium) is applied in a challenge or with a physical stress test (exposure to air, hypoxia, or temperature changes)? Could you briefly discuss these points?

Material and methods

This section is well prepared.

Normality and homoscedasticity of the or qPCR data were tested? Please provide this information.

Conclusions

This section is well prepared.

Reviewer 2 Report

This article “Genome-wide identification, characterization and expression analysis of toll-like receptors in marbled rockfish (Sebastiscus marmoratus)” by Yuan Zhang and co-authors described the genome coding sequences of marbled rockfish for 65 bioinformatics analysis of the marbled rockfish TLR gene family. The article can be recommended for publication in IJMS after approximate modification. Please address the following comments and follow the changes.

(1) Please change the title to “Genome-wide Identification, Characterization and Expression Analysis of Toll-like Receptors in Marbled Rockfish (Sebastiscus marmoratus)”.

(2) In line 11, rephrase

 TLR (toll-like receptor) to “Toll-like receptor (TLR)”

(3) A mechanistic TLR pathway diagram is needed in the introduction for the clear understanding of the working mechanism of TLRs.

(4) Abbreviations qRT-PCR, poly(I:C), LPS, LRRs etc. needs to spell out.

(5) Add more literature support to make an introductory statements, recent work based on TLRs (a) Int. J. Mol. Sci. 2022, 23, 7160 (b) Acc. Chem. Res. 2020, 53, 1046–1055 etc.

(5) In line 45, is there any particular reason for writing TLRs like “tlr1, tlr2, tlr3, tlr5, tlr7, tlr8, tlr9 and 45 tlr13”? It’s very confusing sometime it’s “TLR” sometime its “tlr”. Please follow one format.

(6) In Line 160, please rephrase “Fig. 7(A) and (B)” to “Fig. 7(a) and (b)” as same as in the Figure 7.
